# Activity-dependent regulation of mitochondrial motility in developing cortical dendrites

**Catia AP Silva[1], Annik Yalnizyan-Carson[2], M Victoria Fernández Busch[1], Mike van Zwieten[1], Matthijs Verhage[3], Christian Lohmann[1,3]***

[1]Department of Synapse and Network Development, Netherlands Institute for Neuroscience, Amsterdam, Netherlands; [2]Department of Biological Sciences, University of Toronto Scarborough, Toronto, Canada; [3]Department of Functional Genomics, Center for Neurogenomics and Cognitive Research, University Amsterdam, Amsterdam, Netherlands

**Abstract** Developing neurons form synapses at a high rate. Synaptic transmission is very energy-demanding and likely requires ATP production by mitochondria nearby. Mitochondria might be targeted to active synapses in young dendrites, but whether such motility regulation mechanisms exist is unclear. We investigated the relationship between mitochondrial motility and neuronal activity in the primary visual cortex of young mice in vivo and in slice cultures. During the first 2 postnatal weeks, mitochondrial motility decreases while the frequency of neuronal activity increases. Global calcium transients do not affect mitochondrial motility. However, individual synaptic transmission events precede local mitochondrial arrest. Pharmacological stimulation of synaptic vesicle release, but not focal glutamate application alone, stops mitochondria, suggesting that an unidentified factor co-released with glutamate is required for mitochondrial arrest. A computational model of synaptic transmission-mediated mitochondrial arrest shows that the developmental increase in synapse number and transmission frequency can contribute substantially to the age-dependent decrease of mitochondrial motility.

**\*For correspondence:**
c.lohmann@nin.knaw.nl

**Competing interest:** The authors declare that no competing interests exist.

## Introduction

Newborns can interact with their environment soon after birth, without any previous experience of sensory input. This ability relies on extensive preparation of the developing nervous system before the onset of sensory experience. Young networks are initially established by molecular guidance cues and refined by activity-driven synaptic plasticity. Before the onset of sensory processing, developing neuronal networks generate neuronal activity spontaneously that strengthens well-targeted synapses and weakens others to prepare the brain for sensory processing. Later, learning adjusts synaptic circuits to the prevalent environmental conditions (*Katz and Shatz, 1996*; *Sengpiel and Kind, 2002*; *Sanes and Yamagata, 2009*; *Kilb et al., 2011*; *Kirkby et al., 2013*; *Leighton and Lohmann, 2016*).

The development of synapses and synaptic transmission are highly energy-demanding processes. A substantial amount of this energy is supplied by mitochondria, the main energy providers in neurons (*Harris et al., 2012*). Imaging experiments showed that neuronal mitochondria can be highly motile in intact tissue (*Misgeld et al., 2007*; *Plucińska and Misgeld, 2016*). For example, mitochondria are generated at the soma and transported to distal dendrites and axons via the microtubule network (*Sheng and Cai, 2012*). This motility allows for energy provision at high-energy-demanding sites, in particular, synapses. Defects in mitochondrial motility have been shown to lead to impaired neuro-transmission, further linking mitochondrial motility and synaptic function (*Sheng and Cai, 2012*). In

addition, previous studies reported that experimentally enhancing neuronal activity (with high extra-cellular potassium, the voltage-gated sodium channel activator veratridine, glutamate, or electrical stimulation) stops mitochondria at synapses, whereas blocking action potential firing using tetrodo-toxin (TTX) increases mitochondrial motility and reduces the number of stationary mitochondria at synapses (*Rintoul et al., 2003*; *Li et al., 2004*; *Chang et al., 2006*; *MacAskill et al., 2009*). MIRO1, a calcium-sensitive protein-linking mitochondria to the microtubule network, can mediate mitochon-drial arrest in dendrites and axons (*MacAskill et al., 2009*; *Wang and Schwarz, 2009*): upon calcium binding, MIRO1 releases mitochondria from motor proteins (kinesins or dyneins), thus interrupting their motility.

In contrast, other evidence suggests that mitochondrial motility in neuronal dendrites is not affected by activity (*Beltran-Parrazal et al., 2006*; *Faits et al., 2016*). In retinal explants, neither spontaneously occurring nor stimulus-evoked activity affect mitochondrial motility (*Faits et al., 2016*). Moreover, mitochondrial motility remains high in an hyperactive retina with immature synapses (*Morrow et al., 2005*; *Tran et al., 2014*; *Faits et al., 2016*). These observations suggest that high mitochondrial motility may not be the consequence of low activity in immature tissue, but rather a characteristic of very young neurons (*Faits et al., 2016*). Thus, activity levels may co-vary with mitochondrial arrest rather than causing it.

To address the role of natural activity patterns in mitochondrial arrest, we investigated here whether spontaneous activity affects mitochondrial motility in the developing visual cortex both in vivo and in organotypic slice cultures. We found that mitochondrial motility decreased over the first 2 postnatal weeks while the frequency of spontaneous activity increased. Global spontaneous calcium transients did not affect mitochondrial motility; however, spontaneous activity at the synaptic level preceded mitochondrial motility arrest and pharmacological stimulation of synaptic vesicle release, but not focal glutamate application alone, was sufficient to stop mitochondrial motility. A computational model of synaptic activity-mediated control of mitochondrial motility suggests that the developmental increase in synapse number and transmission frequency contributes substantially to the age-dependent decrease of mitochondrial motility.

## Results

We investigated the relationship between spontaneous activity and mitochondrial motility in vivo and in organotypic slice cultures of the developing mouse primary visual cortex during the second postnatal week before eye opening at postnatal day (P) 14 (*Figure 1A*). We used in utero electropo-ration at embryonic day 16.5 to express the calcium indicator GCaMP6s and mitochondrial-DsRed in pyramidal neurons of layer II/III (*Figure 1B–C*). Time-lapse recordings were performed to reveal the spatio-temporal relationship between mitochondrial motility and calcium signaling in developing dendrites (*Figure 1D–E*).

Previous studies reported that neuronal activity and calcium signaling reduce mitochondrial motility in dendrites in vitro (*Li et al., 2004*; *Chang et al., 2006*), but this idea has not been tested in vivo. Therefore, we first investigated the interaction between spontaneous network activity and mitochon-drial motility in neonatal mice. Overall, we observed an anti-correlation between the frequency of spontaneous global calcium transients and the percentage of moving mitochondria in awake (unanes-thetized) animals (*Figure 2A*). Upon closer inspection, it became clear that these parameters were linked systematically to the age of the animal: in older animals (≥ P8) activity levels were consis-tently higher and mitochondrial motility was low (*Figure 2A–D*). We observed a similar relationship between neuronal activity, mitochondrial motility, and age in anesthetized mice (0.8% isoflurane, *Figure 2—figure supplement 1A-D*). Therefore, we combined both groups for the analyses shown below (*Figure 2E–G*).

Since overall calcium signaling correlated with mitochondrial motility, we asked whether neuronal activity could directly affect mitochondrial motility. First, we replicated previous experiments performed in cell cultures (DIV14–17) that showed an increase of mitochondrial motility after blocking action potential firing (*Li et al., 2004*; *Chang et al., 2006*). Application of the sodium channel blocker TTX (2 μM) to the surface of the brain (P5–12) abolished global calcium transients (*Figure 2E*) and, as expected, led to a significant increase in mitochondrial motility (*Figure 2F*, see also Materials and methods for an extended discussion on statistics). Furthermore, the effect of TTX on mitochondrial

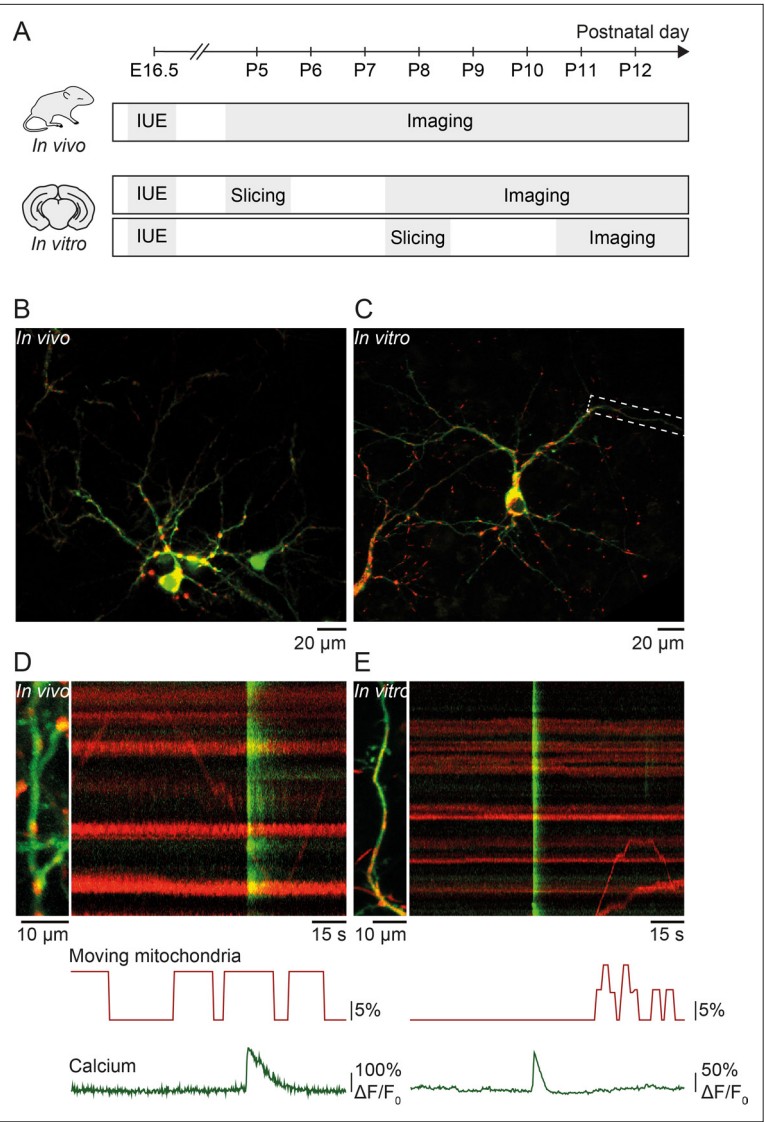

**Figure 1.** Simultaneous imaging of dendritic calcium transients and mitochondrial motility in vitro and in vivo. (**A**) Timeline of in vivo and in vitro experiments: in utero electroporation (IUE) was performed at embryonic day (E) 16.5 to deliver GCaMP6s (calcium indicator) and Mito-DsRed (mitochondrial marker) to pyramidal neurons of layer II/III in the visual cortex. In vivo experiments: acute imaging of transfected dendrites in pups between postnatal day (P) 5 and P12 using a two-photon microscope. In vitro experiments: imaging of transfected dendrites using a confocal microscope in organotypic cortical slices cultured for 3–7 days after slice preparation from P5 or P8 pups. (**B**) GCaMP6- and Mito-DsRed-expressing layer II/III pyramidal neurons in vivo (P16). (**C**) GCaMP6- and Mito-DsRed-expressing layer II/III pyramidal neurons in vitro (P5 + DIV4). (**D**) Dendrite of layer II/III pyramidal neuron in vivo and kymograph (right) representing dendritic calcium transients (green) as well as motile and stationary mitochondria (red). Immobile mitochondria appear as horizontal lines (no change in position over time) and moving mitochondria as diagonal lines. Below, graphic representation of the percentage of moving mitochondria and global calcium transients. The percentage of moving mitochondria was calculated as the number of moving mitochondria over the total number of mitochondria, binned for every second. The mean percentage of moving mitochondria across the duration of this recording was 8.5%. Vertical green lines show spontaneously occurring global calcium transients, most likely resulting from back-propagating action potentials. (**E**) Dendrite of the layer II/III pyramidal neuron shown in C. The mean percentage of moving mitochondria across the duration of the recording was 2.4%.

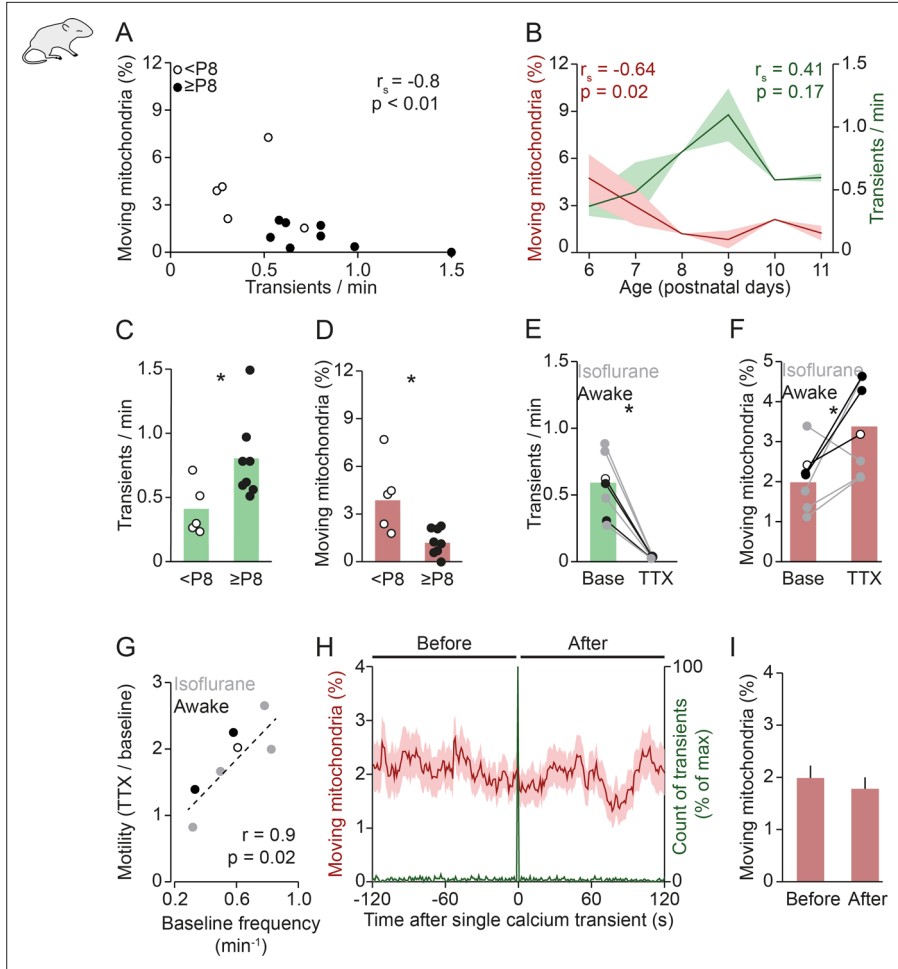

**Figure 2.** Mitochondrial motility and spontaneous activity are anti-correlated during in vivo early postnatal development. (**A**) Anti-correlation between the frequency of spontaneous global calcium transients and the percentage of moving mitochondria in imaging experiments of awake pups (n = 13 pups, Spearman's rank correlation; 1192 mitochondria in 131 dendrites). (**B**) The frequency of spontaneous global calcium transients increased until postnatal day (P) 9 (but not significantly for the entire age range, Spearman's rank correlation) and the percentage of moving mitochondria decreased over P6–11 in vivo (Spearman's rank correlation). (**C-D**) When comparing awake animals younger than P8 to P8 and older, the frequency of spontaneous global calcium transients increased (t-test, n = 5 vs. n = 8, p = 0.02) and the percentage of moving mitochondria decreased (t-test, n = 5 vs. n = 8, p = 0.045). (**E-F**) Application of tetrodotoxin (TTX, 2 µM) on the surface of the cortex (n = 7 pups, 1625 mitochondria in 160 dendrites) completely abolished spontaneously occurring global calcium transients (paired t-test, p = 6*10$^{-4}$) and increased the percentage of moving mitochondria (paired t-test, p = 0.035). (**G**) Higher baseline frequency of spontaneous global calcium transients was correlated with a larger effect of TTX on the percentage of moving mitochondria (n = 7 pups, Pearson correlation, r = 0.85, p = 0.015). (**H-I**): Mean mitochondrial motility time-locked to the onset of single global calcium transients. The percentage of moving mitochondria did not change significantly between the 2 minutes before and after spontaneously occurring global calcium transients in awake animals (n = 136 transients, paired t-test, p = 0.33).

The online version of this article includes the following source data and figure supplement(s) for figure 2:

**Source data 1.** Source data for *Figure 2A-D*.

**Source data 2.** Source data for *Figure 2E-G*.

**Source data 3.** Source data for *Figure 2H*.

**Source data 4.** Source data for *Figure 2I*.

**Figure supplement 1.** Relationship between neuronal activity, mitochondrial motility, and age in vivo.

**Figure supplement 1—source data 1.** Source data for *Figure 2—figure supplement 1A-D*.

*Figure 2 continued on next page*

*Figure 2 continued*

**Figure supplement 1—source data 2.** Source data for *Figure 2—figure supplement 1E*.

**Figure supplement 1—source data 3.** Source data for *Figure 2—figure supplement 1F*.

**Figure supplement 2.** Synaptic calcium transients in vivo.

motility was highly proportional to the frequency of baseline activity ($r^2 = 0.81$; *Figure 2G*), suggesting that natural patterns of neuronal activity efficiently constrain mitochondrial motility.

We then examined whether spontaneously occurring single global calcium transients affected mitochondrial motility. We compared mitochondrial motility before and after global calcium transients across all recordings by aligning the occurrence of global calcium transients in time and plotting the percentage of moving mitochondria around this time point (*Figure 2H*). We found that spontaneous global calcium transients did not precede a change in mitochondrial motility (*Figure 2H–I*) or mitochondrial speed (*Figure 2—figure supplement 1E, F*). Together, these experiments showed that while neuronal activity modulated mitochondrial motility, global calcium transients – most likely reflecting single back-propagating action potentials and bursts of back-propagating action potentials – were ineffective in doing so. We therefore speculated that synaptic transmission, rather than postsynaptic action potential firing, might regulate mitochondrial motility. To address this possibility we aimed at analyzing the relationship between synaptic activity and mitochondrial motility. Our in vivo recordings showed transmission events at individual synapses (*Figure 2—figure supplement 2*), but we detected these events too rarely to quantify any possible effect of synaptic activity on mitochondrial motility.

Therefore, we moved to organotypic slice culture preparations, which allow higher signal-to-noise ratio imaging and more stable recordings to investigate the role of transmission at synapses. We obtained cortical slices at P5 or P8 and kept them in culture for at least 3 days before imaging. Slices obtained from older animals showed a trend toward higher spontaneous activity levels (*Figure 3A*, *Figure 3—figure supplement 1A, B*) and significantly lower mitochondrial motility than slices obtained from younger animals (*Figure 3B*, *Figure 2—figure supplement 1A, B*). As in vivo, spontaneous global calcium transients did not precede changes in mitochondrial motility (*Figure 3C–D*, *Figure 3—figure supplement 1C, D*) or speed (*Figure 3—figure supplement 1E, F*). Thus, mitochondrial motility and its independence of spontaneous global calcium signaling were maintained in slice cultures (*Figure 3—figure supplement 1A, B*). Together, we reproduced our in vivo observations on mitochondrial motility in slice cultures and, thus, found them suitable to investigate the role of synaptic activity in regulating mitochondrial motility.

In slice cultures, visual cortex layer II/III neurons frequently showed spontaneous calcium transients in spines representing synaptic transmission events at excitatory synapses, as shown previously in the developing visual cortex and hippocampus (*Kleindienst et al., 2011*; *Winnubst et al., 2015*; *Niculescu et al., 2018*). In nine cells (P5 + 3–7 DIV), we identified 157 spines of which 71 (45%) showed spontaneous synaptic calcium transients (376 transients). We asked whether synaptic activity affected the motility of passing mitochondria. We observed that mitochondria typically passed by inactive synapses (*Figure 3E*), but frequently halted when they reached a synapse that had just been active (*Figure 3F*). Therefore, we specifically determined whether synaptic calcium transients affected the likelihood that incoming mitochondria stopped at or passed by synapses. To quantify this effect, we compared the percentage of approaching mitochondria that stopped at a synapse before and after the occurrence of a synaptic calcium transient (*Figure 3G–H*). When we compared the percentage of stopping mitochondria during a 120 s interval before a single synaptic calcium transient occurred with an interval of the same duration after that calcium transient, we found that the percentage of stopping mitochondria increased significantly after the transient (*Figure 3I*). To answer whether the observed effect size (the difference between the arrest rates before and after a local calcium transient) was likely to occur by chance or not, we performed a bootstrap analysis where we randomized the time points of synaptic calcium transients in our recordings and determined the resulting effect size for a total of 1000 runs. We found that the observed effect size was above the 95 percentile of the randomized effect size distribution (*Figure 3J*) demonstrating that this effect was unlikely to be observed by chance.

Next, we quantified the effect size for different distances from the synapse and different time intervals after a synaptic calcium transient and found that mitochondrial arrest was most prevalent

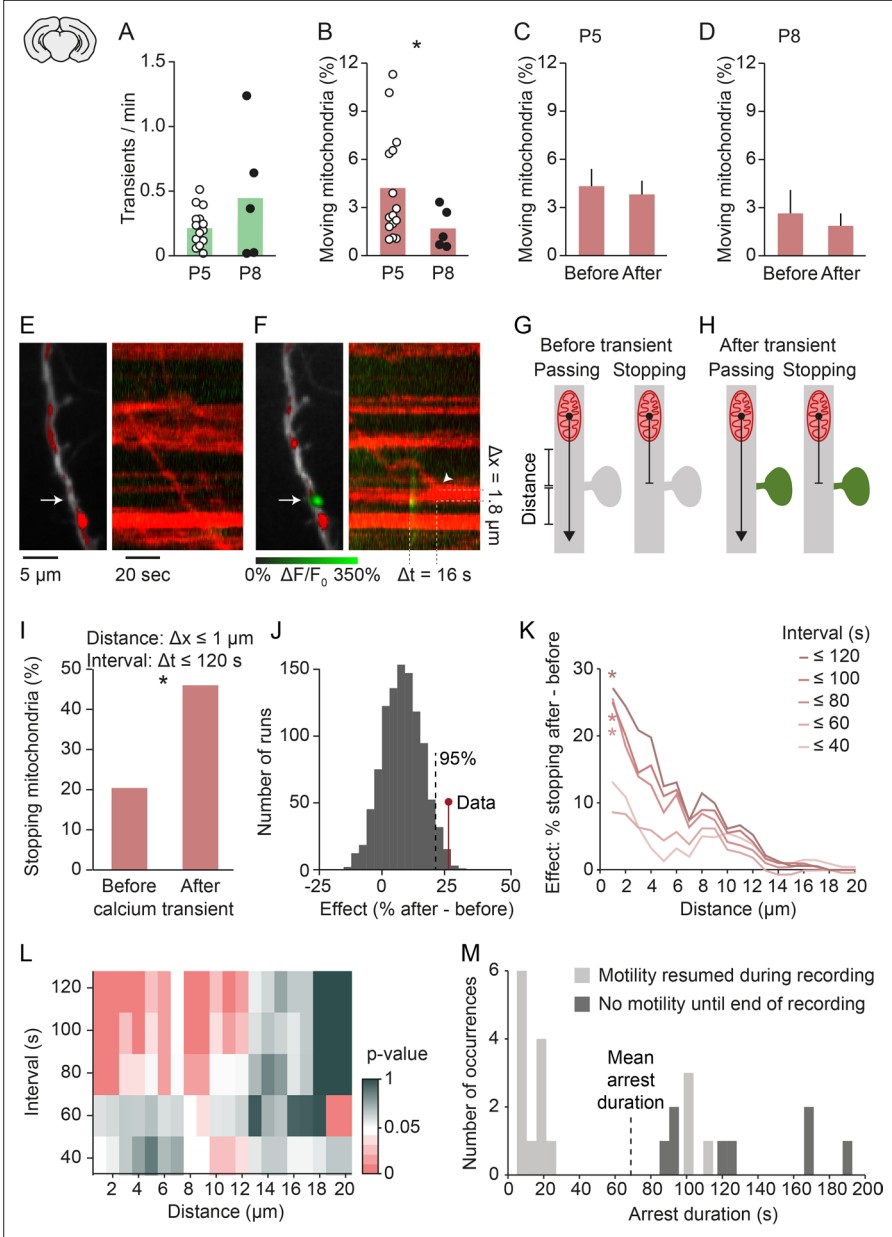

**Figure 3.** Mitochondria stop at synapses after synaptic transmission events. (**A-B**) Frequency of global calcium transients and mitochondrial motility in slices obtained from postnatal day (P) 5 and P8 pups. The frequency of spontaneous global calcium transients did not change significantly (n = 15 vs. n = 5 cells, Student's t-test, p = 0.37). The percentage of moving mitochondria was significantly decreased in slices from older animals (n = 15 [252 mitochondria] vs. n = 5 [85 mitochondria], Student's t-test, p = 0.02), similar to the in vivo results. (**C-D**) The percentage of moving mitochondria did not change significantly between the 2 min before and after spontaneously occurring calcium transients in P5 (n = 158 transients, paired t-test, p = 0.07) or P8 slices (paired t-test, n = 101 transients, p = 0.3). (**E**) Dendritic segment and kymograph showing a mitochondrion approaching and passing an inactive synapse (arrow). (**F**) Same dendritic segment as in A. A mitochondrion arrived near the same synapse (arrow) after a synaptic calcium transient occurred and stopped within its vicinity (Δx: distance to synapse, Δt: time after synaptic calcium transient). (**G-H**) Mitochondria moving toward a synapse can show one of two behaviors: they may continue moving (left) or stop near the synapse (right). We compared the percentage of approaching mitochondria that stopped within a specific distance range before individual synaptic calcium transients occurred (**G**) with that of mitochondria that reached a synapse after a transient (**H**) within a specific time interval. (**I**) There was a significant increase in the percentage of stopping mitochondria after a single local calcium transient occurred (distance ≤ 1 μm; interval ≤ 120 s; *p = $6*10^{-5}$, chi-squared test). (**J**) We compared the

*Figure 3 continued on next page*

*Figure 3 continued*

effect size of mitochondrial arrest at active synapses to a distribution generated by shuffling the time points at which synaptic calcium transients occurred (1000 runs). The observed effect size was within the top 5 percentile of those generated from shuffled data for distances ≤ 1 µm and intervals ≤ 120 s. (**K**) Quantitative estimation of the spatio-temporal characteristics of mitochondrial arrest (chi-squared test for each distance/interval pair Bonferroni-corrected; distance ≤ 1 µm; interval ≤ 80 s, p = 0.0035; interval ≤ 100 s, p = 0.0016; interval ≤ 120 s, p = 0.0025). (**L**) Matrix showing the individual chi-squared test p-values from each distance/interval pair. Roughly, p < 0.05 for intervals between 80 and 120 s and distances of up to 5 µm. (Number of observations for K,L: see *Figure 3—figure supplement 2B*.) (**M**) Distribution of mitochondrial arrest durations after single spontaneous synaptic events. Shown in dark gray are underestimated durations for data points where mitochondria remained immotile until the end of the recording.

The online version of this article includes the following source data and figure supplement(s) for figure 3:

**Source data 1.** Source data for *Figure 3A,B*.

**Source data 2.** Source data for *Figure 3C*.

**Source data 3.** Source data for *Figure 3D*.

**Source data 4.** Source data for *Figure 3I*.

**Source data 5.** Source data for *Figure 3J*.

**Source data 6.** Source data for *Figure 3K*.

**Source data 7.** Source data for *Figure 3L*.

**Source data 8.** Source data for *Figure 3M*.

**Figure supplement 1.** Relationship between neuronal activity, mitochondrial motility, and age in organotypic slice cultures.

**Figure supplement 1—source data 1.** Source data for *Figure 3—figure supplement 1A,B*.

**Figure supplement 1—source data 2.** Source data for *Figure 3—figure supplement 1C*.

**Figure supplement 1—source data 3.** Source data for *Figure 3—figure supplement 1D*.

**Figure supplement 1—source data 4.** Source data for *Figure 3—figure supplement 1E*.

**Figure supplement 1—source data 5.** Source data for *Figure 3—figure supplement 1F*.

**Figure supplement 2.** Number of observations for mitochondrial arrest at individual synapses.

**Figure supplement 2—source data 1.** Source data for *Figure 3—figure supplement 2A*.

**Figure supplement 2—source data 2.** Source data for *Figure 3—figure supplement 2B*.

within distances of up to 5 µm around a synapse and for intervals of 80–120 s after the synaptic event (*Figure 3K and L*). On average, synaptic activity was associated with an interruption of mitochondrial movement for about 1 min (68 s; *Figure 3M*). However, this number underestimated the duration of arrest, since one-third of the stopping mitochondria were still immobile at the end of a recording (mean 131 s at a recording duration of 350 s) preventing an exact estimate of the time point when they started moving again. Together, our observations at individual synapses suggested that spontaneous synaptic transmission can capture moving mitochondria in postsynaptic dendrites.

To address the potential mechanism of mitochondrial arrest at active synapses, we first tested whether membrane depolarization leads to mitochondrial arrest. Consistent with previous studies (*Li et al., 2004*; *MacAskill et al., 2009*; *Faits et al., 2016*), we found that an increase of extracellular potassium to 50 mM decreased mitochondrial motility by approximately 50% (*Figure 4A and B*). This result demonstrated that long-lasting depolarization arrests mitochondria. However, our finding that global calcium transients, which are most likely the consequence of depolarization-induced opening of voltage-gated calcium channels, suggest that depolarization alone is insufficient to stop mitochondria. To test whether synaptic transmission is sufficient to interrupt mitochondrial motility and whether this effect is dependent or independent of action potential firing, we pharmacologically triggered synaptic release while action potential generation was prevented with TTX (*Figure 3C–E*). After three baseline recordings we applied TTX (1 µM), which blocked all global calcium transients as expected. Next, we stimulated release of synaptic vesicles by applying latrotoxin (*Deak et al., 2009*) to the bath. The molecular mechanism of latrotoxin-induced transmitter release is unknown. Nevertheless, at the concentration used here (1 nM), latrotoxin specifically triggers synaptic vesicle release through activation of its receptors latrophilin and neurexin that are located at presynaptic terminals

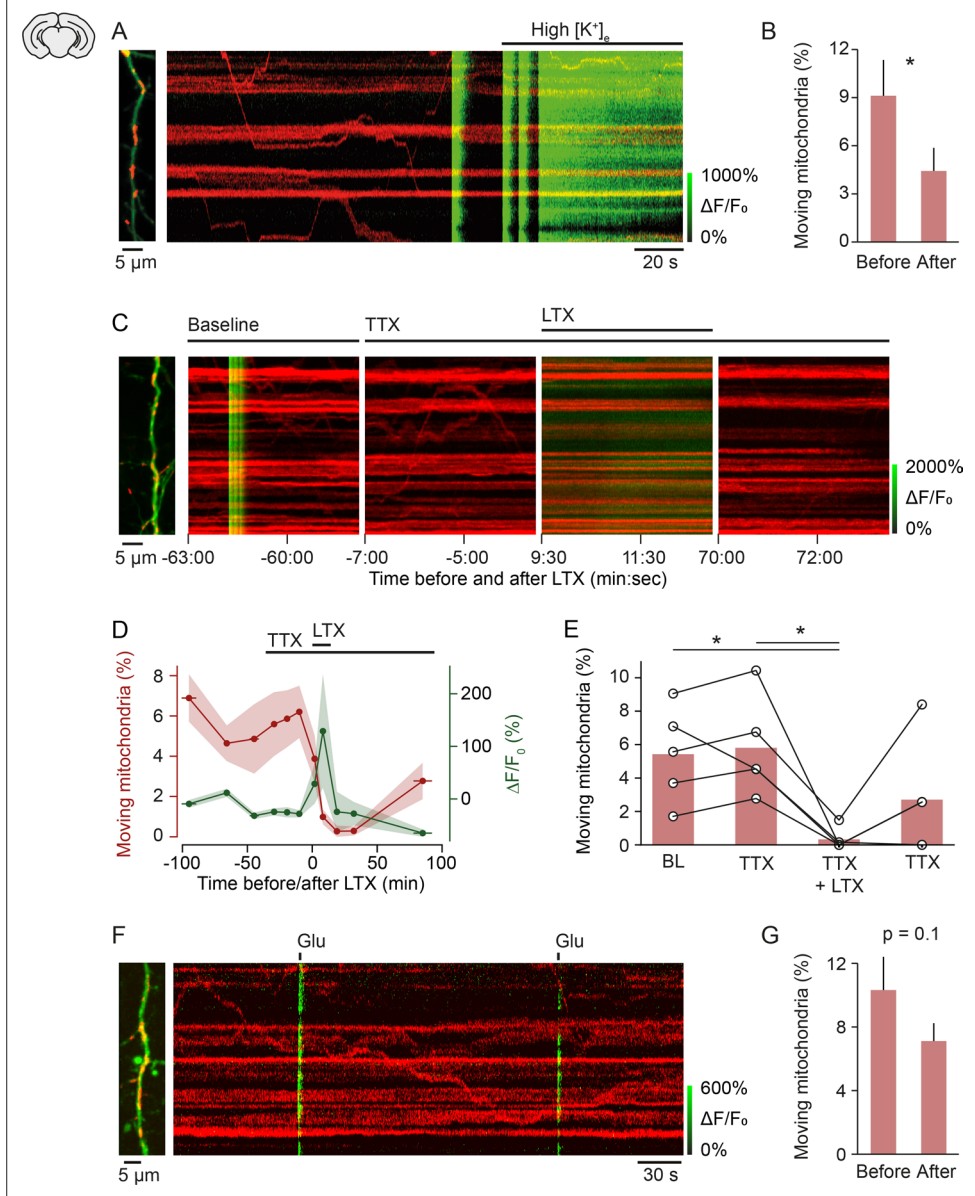

**Figure 4.** Mechanism of activity-induced mitochondrial arrest. (**A, B**) Perfusing layer II/III pyramidal neurons (postnatal day [P] 5 + 3–7 DIV) with high-potassium medium [50 mM] triggered a massive influx of calcium and significantly reduced mitochondrial motility within 2 min (n = 9 cells; 107 mitochondria, paired t-test, p = 0.04). (**C-E**) Stimulating synaptic vesicle release with latrotoxin (LTX) interrupted mitochondrial motility entirely. (**C**) Example kymographs from recordings during baseline, in the presence of tetrodotoxin (TTX), TTX and LTX, and after washout of LTX. Basal calcium levels were elevated and mitochondrial motility was absent during the presence of LTX. (**D**) Averaged time course of mitochondrial motility and GCaMP6 $\Delta F/F_0$ for the duration of the experiments. Shaded areas and horizontal bars indicate SEMs of values and time points, respectively. (**E**) Percentage of moving mitochondria across all conditions (p = 0.0058, repeated measures ANOVA, *p = 0.028 (baseline vs. LTX + TTX), *p = 0.022 (TTX vs. TTX + LTX), post hoc t-test with Bonferroni multi-measures correction, n = 5 cells, 92 mitochondria). (**F-G**) Triggering calcium transients with focal application of glutamate (100 µM) in the presence of TTX did not affect mitochondrial motility significantly (P5 + 3–7 DIV, n = 74 transients from 13 cells, 146 mitochondria, paired t-test, before vs. after, 10.32 ± 2.09 vs. 7.12 ± 1.12, p = 0.1).

The online version of this article includes the following source data for figure 4:

**Source data 1.** Source data for *Figure 4B*.

**Source data 2.** Source data for *Figure 4D,E*.

**Source data 3.** Source data for *Figure 4G*.

(*Valtorta et al., 1984*; *Matteoli et al., 1988*; *Südhof, 2001*). After application of latrotoxin to the bath, the intracellular calcium concentration increased within a few minutes and mitochondrial motility was either entirely suppressed or largely inhibited (*Figure 3C–E*). Mitochondrial motility recovered only partly after around 1 hr, whereas calcium levels returned to baseline levels within 15–20 min.

These experiments indicated that single synaptic transmission events have the capacity to stop mitochondria for 1 to a few minutes and that massive synaptic activation interrupts mitochondrial motility almost entirely for periods of tens of minutes. Finally, we asked whether the transmitter glutamate is responsible for presynaptic release-mediated mitochondrial arrest. We applied single puffs of glutamate (100 µM) to individual dendrites using a pico-spritzer. Focal glutamate delivery triggered calcium increases in the dendrite extending 7–59 µm (27 ± 14 µm; mean ± standard deviation). We analyzed mitochondrial motility before and after glutamate application in the dendritic stretch that responded with a calcium rise. We found that mitochondrial motility was slightly, but not significantly, reduced after glutamate puffs (*Figure 4F and G*), demonstrating that glutamate is not sufficient to cause vesicle release-mediated mitochondrial arrest.

While the factor that mediates mitochondrial arrest remains unknown, our experiments showed that synaptic vesicle release interrupts mitochondrial transport locally. Since synaptic density (*De Felipe et al., 1997*) as well as network activity (*Rochefort et al., 2009*) and thus synaptic vesicle release increase dramatically in the cortex during the second postnatal week, we hypothesized that the temporary recruitment of mitochondria to synapses by spontaneous synaptic activity could contribute to the overall decrease in mitochondrial motility we observed during in vivo development.

To investigate this idea, we designed a computational model for estimating mitochondrial motility in a developing dendrite at different synaptic input frequencies. Since we established a lower bound for the mean duration of immobilization of approximately 70 s, we modeled the effect of synaptic inputs on mitochondrial motility for mean arrest durations of 1–5 min using Gaussian distributions (µ = 1–5 min, σ = 2.5 min). We found that the distributions for 1–3 min were comparable with our observed duration distributions (*Figure 5A*, *Figure 3M*). The model showed that changes in synaptic activity could

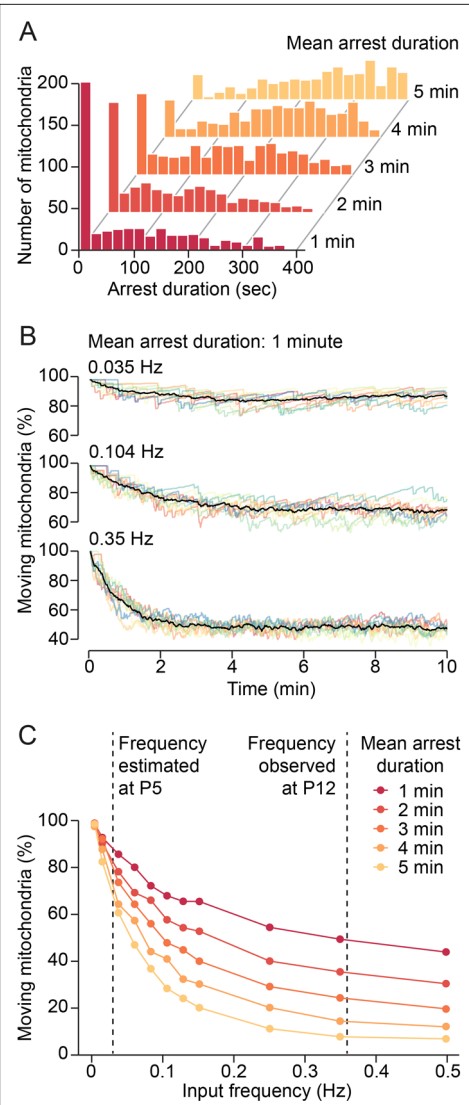

**Figure 5.** Model of synaptic input-mediated modulation of mitochondrial motility. (**A**) Distribution of mitochondrial arrest durations generated by the model for mean durations of 1–5 min. (**B**) Changes of mitochondrial motility after onset of simulated synaptic inputs for a mean arrest duration of 1 min. Input frequencies are given as total synaptic inputs along a 100 µm stretch of dendrite. Low input frequencies hardly changed overall mitochondrial motility. Higher input frequencies reduced motility substantially. Steady state was reached after a few minutes. Colored lines: individual simulations, black lines: average of 10 simulations. (**C**) Relationship between synaptic input frequency and mitochondrial motility for different arrest durations at steady state. The expected increase of synaptic activity from postnatal day (P) 5 to P12 reduced mitochondrial motility by 30–60%, depending on the actual duration of mitochondrial arrest after synaptic transmission.

affect mitochondrial motility critically: while low input frequencies (e.g. 0.035 Hz in a 100 μm dendritic segment) reduced mitochondrial motility only marginally from the default state (approximately 10%), higher input frequencies showed clear effects (*Figure 5B*). For example, at 0.35 Hz, mitochondrial motility was reduced by approximately 50% at steady state. We determined mitochondrial motility for increasing input frequencies and different durations of mitochondrial arrest (*Figure 5C*). To determine the putative effect of an increase in synaptic activity between the first and the second postnatal week on mitochondrial motility, we estimated the frequency of synaptic inputs received by a stretch of dendrite of 100 μm length. In our previous in vivo study, we found a mean density of 36 active synapses per 100 μm dendrite in visual cortex layer II/III pyramidal neurons at the end of the second postnatal week (P10–15) and transmission occurred 0.6 times per minute at each synapse (*Winnubst et al., 2015*). We estimated, therefore, that a 100 μm dendrite receives synaptic inputs at a frequency of approximately 0.36 Hz. Anatomical studies showed a 5–10 times increase of synaptic density in the developing sensory cortex between P5 and P11 (*De Felipe et al., 1997*) and we found here that neuronal activity doubled from the first to the second postnatal week (*Figure 2C*). Assuming that release probabilities do not change dramatically during this period, synaptic input frequencies should be about 0.02–0.04 Hz at P5. Similarly, we find in in vivo patch-clamp and calcium imaging experiments a 16 times increase in synaptic transmission events in dendrites of V1 layer II/III pyramidal cells from P8 to P13 (AH Leighton et al. 2021, in preparation). As *Figure 5C* shows, an increase in synaptic input frequency in this range would reduce mitochondrial motility by 30–60% depending on the actual arrest times at synapses. Since we observed a 70% decrease in mitochondrial motility between the first and second postnatal week, our model showed that temporary immobilization of mitochondrial motility through synaptic signaling can mediate a large proportion of this effect. Together, our data suggest a primary role of synaptic vesicle release, but not action potential firing, in the reduction of mitochondrial motility with dendrite maturation.

## Discussion

Mitochondrial motility and positioning are fundamental for axon and dendrite development and synaptic plasticity (*Courchet et al., 2013*; *Kimura and Murakami, 2014*; *Fukumitsu et al., 2015*; *López-Doménech et al., 2016*; *Vaccaro et al., 2017*; *Divakaruni et al., 2018*). Moreover, mitochondrial dynamics are affected in many neurological disorders (*Chen and Chan, 2009*; *Deheshi et al., 2013*; *Misgeld and Schwarz, 2017*). Thus, regulation of mitochondrial motility is important for neuronal function; however, to what degree neuronal activity determines mitochondrial motility in intact neuronal circuits has been unclear. By directly observing mitochondrial motility and neuronal activity simultaneously in developing dendrites, we provide evidence that synaptic vesicle release, but not postsynaptic action potential firing, constrains mitochondrial motility and stabilizes mitochondria with increasing age.

Imaging neuronal activity and dendritic mitochondria in vivo demonstrated a dramatic motility reduction of visual cortex layer II/III pyramidal cells during the second postnatal week. Motility reduction progressed in parallel with a strong increase in overall neuronal activity. Given that dendritic mitochondria in retinal explants and axonal mitochondria in the visual cortex show similar decreases in motility (*Chang and Reynolds, 2006*; *Faits et al., 2016*; *Lewis et al., 2016*; *Smit-Rigter et al., 2016*), our results confirm a general progression towards more stationary mitochondria in intact tissue.

Since there have been conflicting reports on the regulation of mitochondrial motility by natural activity patterns, we set out to investigate their relationship directly. We found that global calcium transients reflecting back-propagating action potentials were unrelated to changes in mitochondrial motility. Considering that manipulations of neuronal action potential firing did trigger changes in mitochondrial motility in the present study and many studies in cell cultures (*Li et al., 2004*; *Chang et al., 2006*; *MacAskill et al., 2009*; *Wang and Schwarz, 2009*; but see: *Beltran-Parrazal et al., 2006*), this finding was surprising at first (but see below).

As global calcium transients appeared to be ineffective, we studied the role of transmission at individual synapses and discovered that the likelihood for a passing mitochondrion to stop at a synapse increased significantly when this synapse was active within 2 min before it arrived. That synaptic activation, but not action potential firing, arrests mitochondrial motility is in fact consistent with the majority of observations of activity-dependent regulation of mitochondrial motility in hippocampal and cortical neurons (*Li et al., 2004*; *Chang et al., 2006*; *MacAskill et al., 2009*; *Wang and Schwarz, 2009*). Since

activity manipulations, such as depolarizing neurons by increasing extracellular potassium, blocking action potential firing with TTX, and electric field stimulation affect both action potential firing and synaptic transmission, it is possible that synaptic release changes, but not firing alone, altered mitochondrial motility in these studies.

How can local synaptic activation stop moving mitochondria when their motility is unaffected by spontaneously occurring global calcium transients? Our observations that pharmacological activation of synaptic vesicle release arrests mitochondria, but that neither spontaneous global activation nor focal glutamate application stops moving mitochondria, suggest that a local factor, either by itself or together with glutamate, mediates mitochondrial arrest. A possible candidate is ATP: it is present in synaptic vesicles and released simultaneously with glutamate at excitatory cortical synapses (*Khakh, 2001*; *Burnstock, 2007*; *Lalo et al., 2016*). ATP receptors of the P2X and P2Y families are expressed in cortical pyramidal cell dendrites (*Guzman and Gerevich, 2016*), trigger postsynaptic depolarizations (*Pankratov et al., 2002*) and calcium increases (*Lalo et al., 1998*; *Lalo and Kostyuk, 1998*), activate CaMKII (*Pougnet et al., 2014*) and mediate synaptic plasticity (*Pankratov et al., 2009*; *Lalo et al., 2016*). Thus, a local factor co-released with glutamate, such as ATP, is most likely required for mitochondrial arrest at active synapses. A co-released factor could, in principle, provide single synapse specificity by generating or – together with glutamate – boosting a local signal that stops mitochondria at active synapses.

Our analysis of changes in mitochondrial motility in relation to spontaneously occurring synaptic transmission events allowed us to determine the spatio-temporal characteristics of this effect quantitatively. We found that mitochondrial arrest was restricted to a segment of dendrite of roughly 5–10 μm distally and proximally from the insertion point of a spine. Mitochondria were stopped when they arrived within 2 min after a synaptic transmission event and remained stationary for 1 min on average. Interestingly, several molecular signaling cascades at the synapse have been described that act on very similar scales in time and space. In particular, several small GTPases become activated within less than a minute after single spine stimulation in short stretches of dendrite (5–10 μm) and stay active for several minutes (e.g. Ras and RhoA; *Harvey et al., 2008*; *Murakoshi et al., 2011*). These and other small GTPases, including DRP-1 and Miro-1, which regulate mitochondrial activity and motility, are controlled by intracellular calcium rises and CaMKII activation (*MacAskill et al., 2009*; *Wang and Schwarz, 2009*; *Fukumitsu et al., 2016*; *Divakaruni et al., 2018*). CaMKII expression increases dramatically in the visual cortex during the second postnatal week (2008 Allen Institute for Brain Science. Allen Developing Mouse Brain Atlas. Available from: https://developingmouse.brain-map.org/). Therefore, synaptic transmission-induced CaMKII phosphorylation requiring, for example, ATP receptor activation may stop mitochondria more frequently with increasing age by activating small GTPases for a few minutes and several micrometers along the dendrite.

To estimate whether mitochondrial arrest through synaptic activity can explain the progressive demobilization of mitochondria in dendrites during development, we employed a computational model. This model indicates that the estimated increase in synaptic activity from the first to the second postnatal week can reduce mitochondrial motility by 30–60%. These numbers are in line with our in vivo observation that blocking synaptic activity with TTX increased mitochondrial motility by 60%. Together, these data show that developmental increases in synaptic activity can explain a large proportion of the motility decrease observed during this period. We speculate that the here described reduction of mitochondrial motility through synaptic activity with increasing age is complemented by a shift in the number of potentially mobile mitochondria toward a pool of stationary mitochondria during development. While for example Miro1 controls temporary mitochondrial arrest, there is no molecular mechanism known for retaining mitochondria permanently at a location in dendrites. Theoretical models suggest that mitochondria are stationary in the absence of Miro1 (*MacAskill et al., 2009*); however, in Miro1 knockout neurons mitochondrial motility is only mildly affected (*Saotome et al., 2008*; *MacAskill et al., 2009*; *López-Doménech et al., 2018*). Furthermore, to our knowledge a reduction in Miro1 expression or function during development has not been reported. Alternatively, increased tethering of mitochondria may reduce the pool of potentially mobile mitochondria with increasing age. For example, myosin V anchors mitochondria (*Pathak et al., 2010*), has been proposed to keep mitochondria in a stationary state (*Schwarz, 2013*; *Misgeld and Schwarz, 2017*), and is enriched in dendrites (*Wang et al., 2008*; *Konietzny et al., 2019*).

The regulation of mitochondrial motility through synaptic activity we describe here may serve developing synapses to efficiently meet their energy demands and calcium handling. In addition, synaptic regulation of mitochondrial trafficking can account to a large degree for the reduction of mitochondrial motility during development and is probably a fundamental process in wiring the developing brain.

# Materials and methods

## Key resources table

| Reagent type (species) or resource | Designation | Source or reference | Identifiers | Additional information |
|---|---|---|---|---|
| Strain, strain background (*Mus musculus*, male/female) | C57Bl/6 J, male and female | Janvier Labs; | | |
| Transfected construct | GCaMP6s (species: *Rattus norvegicus*) | Add gene plasmid 40753; Douglas Kim | RRID:Addgene_40753 | Cloned into pCAG vector |
| Transfected construct | Mitochondrial-DsRed (species: *Homo sapiens*) | Gift from Thomas Misgeld | | Mitochondrial targeting sequence from subunit VIII of human cytochrome c oxidase causing mitochondrial localization as previously described; *Rizzuto et al., 1995*; *Li et al., 2004*; *MacAskill et al., 2009* Cloned into pCAG vector |
| Chemical compound, drug | TTX | 1078, Bio-Techne, Minneapolis, MN | | |
| Chemical compound, drug | LTX | ALX-630–027 C040, Enzo Life Sciences b.v., Farmingdale, NY | | |
| Chemical compound, drug | Glutamate | G1626, Sigma | | |
| Software, algorithm | MitoMotil | This study | | https://github.com/annikc/MitoMotil (copy archived at swh:1:rev:a4cfb2b4fd66579f63ea5a150a0f9b1b21b89a83, *Yalnizyan-Carson, 2021*) |
| Software, algorithm | MATLAB | The MathWorks | | https://mathworks.com |
| Software, algorithm | NormCorre | Flatiron Institute, Simons Foundation | | https://github.com/flatironinstitute/NoRMCorre, *Pnevmatikakis and Giovannucci, 2021* |
| Software, algorithm | Python | Python Software Foundation | | https://www.python.org/ |
| Software, algorithm | Elephant library | Human Brain Project | | https://elephant.readthedocs.io/en/latest/ |

## Plasmids

To investigate the relationship between neuronal activity and mitochondria, we used the genetically encoded calcium indicator GCaMP6s (Addgene plasmid 40753; Douglas Kim) in combination with mitochondrial-DsRed (mitochondrial targeting sequence from subunit VIII of human cytochrome *c* oxidase causing mitochondrial localization as previously described; *Rizzuto et al., 1995*; *Li et al., 2004*; *MacAskill et al., 2009*). These plasmids were cloned into pCAGGS, to enable delivery to neurons via in utero electroporation.

## Animals and in utero electroporation

All experimental procedures were approved by the institutional animal care and use committee of the Royal Netherlands Academy of Arts and Sciences. To sparsely deliver the plasmids of interest to pyramidal neurons of layer II/III of the visual cortex, pregnant C57Bl/6 J female mice at 16.5 days gestation underwent in utero electroporation surgery. Pregnant females were anesthetized using 3% isoflurane mixed with 1 l/min oxygen and kept under anesthesia with 1.5–2% isoflurane. A midline incision was made and uterine horns were exposed. Plasmid DNA (mitochondrial-DsRed: 0.1 µg/µl,

GCaMP6s: 2 µg/µl) was dissolved in 10 mM Tris and 0.05% Fast Green. Approximately 1 µl of this mixture was injected through a pulled capillary pipette in the lateral ventricle of each embryo using a picospritzer (PLI-100, BTX Harvard Apparatus, Holliston, MA). A custom-made square wave isolated pulse generator (voltage of 50 V, 5 pulses, pulse width 50 ms, and 150 ms interval) was used for electroporation. After electroporation, the uterine horns were carefully placed back in the abdomen cavity and the abdomen was sutured. During the surgery, embryos were kept moist with warm saline and the mothers were kept warm using a euthermic pad. Pregnant females were allowed to recover after Lidocaine ointment was applied on the wound for local analgesia and Metacam (1 mg/kg s.c.) was administered for post-operative analgesia. Once the pups were born they were checked before P2 for expression and targeting of V1.

### Organotypic slice cultures

Organotypic slice cultures of transfected visual cortex were prepared as follows: at P5 or P8, animals were decapitated quickly, and brains were placed in ice-cold Gey's balanced salt solution under sterile conditions. Coronal slices (400 µm for P5 and 250 µm for P8) were cut using a tissue chopper (McIllwain) and incubated with serum-containing medium on Millicell culture inserts (Millipore, Merck, New York, NY). Slices were kept in culture for 3–7 days before imaging.

### Confocal microscopy of organotypic slice cultures

For confocal imaging, slices were excised from their membrane supports and placed in a flow-through chamber. Slices were continuously perfused with heated (35 °C) Hank's Balanced Salt Solution (HBSS, Fisher Scientific, Waltham, MA, supplemented with in mM: 4.2 NaHCO3, 2.6 CaCl2, 0.1 Trolox). Slices were imaged on a SP5 Leica confocal microscope with a 63 × objective (0.9 NA, Leica, Wetzlar, Germany). For imaging we selected neurons that showed the following characteristics: soma localized in upper layer II, apical dendrite pointing to layer I, low basal GCaMP6s fluorescence as well as long and dim mitochondria. Preference was given to isolated cells, to minimize background fluorescence. Apical dendrites (at least 50 µm from the soma) were imaged using an argon laser at 488 nm and power levels between 0.3% and 1%. Time-lapse image stacks (up to six optical sections, 1.2 µm z-spacing), at 0.23 µm per pixel, 350 ms per stack were collected for 350 s, every 10 min, for a total of 10 times per cell. We observed no changes in fluorescence intensity, cell activity levels, or mitochondrial motility levels with time under these conditions. At the end of the experiment, low magnification image stacks (0.23 µm pixel size and 1 µm z-spacing) were collected to localize the recorded dendrite within the dendritic arborization.

### In vivo two-photon microscopy

For in vivo imaging, transfected neonatal mice (P5–12) were pre-anesthetized using 3% isoflurane mixed with 1 l/min oxygen and kept under anesthesia with 1–2% isoflurane. A head bar with an opening (4 mm Ø) was attached to the skull above the visual cortex (0–2 mm rostral from lambda and 0–2 mm lateral from the midline) with superglue (Henkel, Düsseldorf, Germany) and dental cement (Heraeus, Hanau, Germany). A small craniotomy above the visual cortex (approximately 1–2 mm Ø) was performed with a needle and forceps and care was taken not to damage the dura mater. The exposed cortical surface was kept moist with cortex buffer (in mM: 125 NaCl, 5 KCl, 10 glucose, 10 HEPES, 2 $MgSO_4$, 2 $CaCl_2$, pH 7.4). For additional stability, a thin layer of 1.5% high electroendosmosis agarose (Biomol, Hamburg, Germany) was applied to the cortical surface. Before imaging, isoflurane was decreased to 0.8% (under anesthesia condition) or 0% (awake condition). A pulsed titanium sapphire laser (Chameleon Vision II, Coherent, Palo Alto, CA) at 900 nm and power up to 30% was used with a 25 × water-immersion objective (1.10 NA, Nikon). Time-lapse image stacks (up to five optical sections, 2 µm z-spacing) were obtained at a pixel size of 0.13–0.17 µm and stack rate of 5–10 Hz. Throughout the entire experiment, physiological parameters such as heartbeat and body temperature were monitored, and temperature was controlled using a heating pad.

### Pharmacological manipulations

High extracellular potassium in vitro: cells were imaged as described above for at least 20 min and then the imaging medium was replaced by one supplemented with KCl to a final concentration of 50 mM. After 20 min of high potassium incubation and imaging, normal medium was restored and

cells were imaged for at least 20 more minutes. Cells did not show dendritic blebbing and in most cells spontaneous activity reappeared, suggesting that they were healthy until the end of the experiment.

To test the role of presynaptic release on mitochondrial motility in dendrites, we imaged the effects of latrotoxin, a synaptic vesicle release stimulator (*Deak et al., 2009*), on mitochondrial motility and calcium levels in slice cultures during the following conditions sequentially: baseline, TTX, TTX+ LTX, and after LTX washout. TTX was applied (1078, 1 µM, Bio-Techne, Minneapolis, MN) through the bath perfusion. Then LTX (ALX-630–027 C040, 1 nM, Enzo Life Sciences b.v., Farmingdale, NY) was added to the bath and the perfusion was stopped for 10 min. Subsequently, perfusion resumed with TTX containing solution.

Focal glutamate application: TTX (1 µM; No. 1078, Bio-Techne, Minneapolis, MN) was applied through the bath perfusion. A glass pipette with a resistance of approximately 4 MΩ containing glutamate (100 µM) in bath solution was inserted into the slice, approximately 50 µm from the dendrite, and glutamate was applied focally with a Picospritzer at 20 psi (PLI-100, BTX Harvard Apparatus, Holliston, MA). Pulse duration was chosen between 1 and 20  ms to evoke local calcium transients. After placing the glutamate-containing pipette, adjusting the pulse duration and a wait period of at least 10 min, 1–3 single pulses were applied during each recording of 350 s duration.

To block action potential firing in vivo, TTX (2 µM) was prepared in cortex buffer and in agarose. After baseline imaging, the agarose was carefully removed from the top of the brain and the TTX solution in cortex buffer was applied to the surface of the brain for 2 min. Then, this cortex buffer was removed and agarose with TTX was applied to the surface of the brain. Imaging continued as previously. This procedure blocked neuronal activity for the entire imaging period, while the pups' physiological parameters did not change.

## Image analysis

All images were processed using ImageJ software. Images were filtered using a median filter (radius one pixel). Maximal intensity projections of image stacks were generated. All stacks recorded at one dendrite were corrected for motion artifacts due to drift as well as aligned with respect to each other using NoRMCorre (*Pnevmatikakis and Giovannucci, 2017*).

From the resulting stacks, two-dimensional projections of time (x-axis) vs. displacement (y-axis) were generated for individual dendrites to examine spontaneous global calcium transients as well as mitochondrial motility. Global calcium transients appeared as vertical lines, as there was an increase in intracellular calcium levels throughout the entire dendrite. Immotile mitochondria appeared as horizontal lines, and mitochondrial motility as diagonal lines. The percentage of moving mitochondria was calculated as the number of moving mitochondria divided by the total number of present mitochondria, for each second.

For the analysis of local calcium transients, $\Delta F/F_0$ images were calculated where $F_0$ was the average fluorescence of the first 200 frames without apparent calcium transients of the first recording for each cell. Custom-made Matlab scripts aided the manual identification of synaptic events: signals had to last for more than the duration of two frames, did not spread from other sites, and were localized to the spine head.

## Statistics

Calcium transients per minute and percent moving mitochondria per 1 s bins are shown in all figures where we compare global calcium transient activity or mitochondrial motility across time or different experimental conditions, respectively. Spearman's rank correlation was used to detect correlations across time. For single comparisons t-tests (two-tailed, paired, or unpaired) and for multiple comparisons repeated measures ANOVA with post hoc t-tests and Bonferroni multi-measure correction were used. Since for the in vivo measurements (*Figure 2*) the initial percentages of moving mitochondria were very low, the observed effect may be susceptible to discretization. To test whether age and TTX do indeed affect mitochondrial motility in vivo, we performed additional analyses. The number of moving and stable mitochondria were counted in 2 min bins (*MacAskill et al., 2009*) and then summed across all recordings for both conditions, respectively, and the resulting contingency tables (see Source Data Tables) were used to perform Fisher's exact test. We found that the number of observed mitochondria moving was significantly decreased in animals that were P8 or older compared to younger animals ($p < 0.00001$) and that TTX increased the number of moving mitochondria significantly ($p = 0.0034$). To

test whether there is a significant relationship between the occurrence of individual synaptic calcium transients and the arrest of mitochondria, we used chi-squared tests and performed a bootstrap analysis as described in the Results section.

## Modeling mitochondrial motility modulation by synaptic inputs (MitoMotil)

The model of mitochondrial motility was written with Python 3.6 (Python Software Foundation). First, a population of mitochondria (n = 500) was generated where each mitochondrion was initialized with a recovery time drawn from a normal distribution. We ran simulations varying recovery time distribution means over 1–5 min ($\sigma$ = 2.5 min in each condition). All mitochondria were in the motile pool at the beginning of the simulation run. Synaptic transmission events were generated from a homogeneous Poisson process (from the Elephant library, https://elephant.readthedocs.io/en/latest/) with synaptic input frequencies ranging from 0.001 to 0.5 Hz. We ran each simulation for 1500 s to allow the percentage of immobile mitochondria to reach steady state. Each synaptic transmission event immobilized a variable number of mitochondria randomly selected from the total pool. The proportion of affected mitochondria was drawn from a normal distribution ($\mu$ = 0.05, $\sigma$ = 0.01). These values were based on our observation that single synaptic transmission events affected approximately 5–10 $\mu$m of a 100 $\mu$m stretch of dendrite. This variable proportion of affected mitochondria was used to calculate the number of mitochondria from the population for immobilization. Affected mitochondria remained immobilized for the duration of the recovery time variable with which they were initialized. If a mitochondrion was already immobilized and selected from the total pool for immobilization by a subsequent event, the immobilization time was extended by the second event.

## Acknowledgements

We thank Ginny Farias, Koen Kole, Thomas Misgeld, and Rajeev Rajendran for critically reading the manuscript. Johan Winnubst, Juliette Cheyne, and Alexandra Leighton for custom-made Matlab scripts and Matthew Self for advice on statistics, Thomas Misgeld for the original mitochondrial-DsRed plasmid, and Christiaan Levelt for the pCAGGS construct. In addition, we thank Christiaan Levelt for teaching us the in utero electroporation surgery.

## Additional information

### Funding

| Funder | Grant reference number | Author |
| --- | --- | --- |
| Nederlandse Organisatie voor Wetenschappelijk Onderzoek | 819.02.017 | Christian Lohmann |
| Nederlandse Organisatie voor Wetenschappelijk Onderzoek | 822.02.006 | Christian Lohmann |
| Nederlandse Organisatie voor Wetenschappelijk Onderzoek | ALWOP.216 | Christian Lohmann |
| Nederlandse Organisatie voor Wetenschappelijk Onderzoek | 865.12.001 | Christian Lohmann |
| Nederlandse Organisatie voor Wetenschappelijk Onderzoek | OCENW.KLEIN.535 | Christian Lohmann |
| Stichting Vrienden van het Herseninstituut | 822.02.006 | Christian Lohmann |

The funders had no role in study design, data collection and interpretation, or the decision to submit the work for publication.

## Author contributions
Catia AP Silva, Conceptualization, Formal analysis, Investigation, Writing – original draft, Writing – review and editing; Annik Yalnizyan-Carson, Formal analysis, Software; M Victoria Fernández Busch, Mike van Zwieten, Investigation; Matthijs Verhage, Conceptualization, Supervision, Writing – review and editing; Christian Lohmann, Conceptualization, Investigation, Project administration, Supervision, Writing – original draft, Writing – review and editing

## Author ORCIDs
Catia AP Silva http://orcid.org/0000-0001-8801-3508
Annik Yalnizyan-Carson http://orcid.org/0000-0002-1664-5327
M Victoria Fernández Busch http://orcid.org/0000-0001-8636-4988
Christian Lohmann http://orcid.org/0000-0002-1780-2419

## Ethics
All experimental procedures were approved by the institutional animal care and use committee of the Royal Netherlands Academy of Arts and Sciences. License number: AVD801002015249.

## Decision letter and Author response
Decision letter https://doi.org/10.7554/eLife.62091.sa1
Author response https://doi.org/10.7554/eLife.62091.sa2

---

# Additional files

## Supplementary files
• Transparent reporting form

## Data availability
All data generated or analyzed during this study are included in the manuscript and supporting files. Source data files have been provided for all figures and figure supplements.

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
