## [Decision Letter]

Thank you for submitting your article "Synaptic inputs, but not action potentials, regulate motility of dendritic mitochondria in the developing visual cortex" for consideration by *eLife*. Your article has been reviewed by 3 peer reviewers, one of whom is a member of our Board of Reviewing editors, and the evaluation has been overseen by Gary Westbrook as the Senior Editor. The reviewers have opted to remain anonymous.

The reviewers have discussed the reviews with one another and the Reviewing Editor has drafted this decision to help you prepare a revised submission. The editors and reviewers have judged that your manuscript is of interest, but as described below additional experiments are required before it could be published.

We would like to draw your attention to changes in our revision policy in response to COVID-19 (https://elifesciences.org/articles/57162). First, because many researchers have temporarily lost access to labs, we will give authors as much time as they need to submit revised manuscripts. We are also offering, if you choose, to post the manuscript to bioRxiv (if it is not already there) along with this decision letter and a formal designation that the manuscript is "in revision at eLife". Please let us know if you would like to pursue this option.

Overview of discussion between Editors and Reviewers:

This study explores the relationship between calcium signaling and mitochondrial motility in dendrites of developing visual cortex neurons. The manuscript provides interesting new insights into the regulation of mitochondrial movements in dendrites. Specifically it explains apparent previous discrepancies in studies on the role of activity in regulating mitochondrial motility by identifying differences between the impact of synaptic activity vs. action potential firing. The experiments appear carefully conducted, the findings are well illustrated, and communicated clearly in the text.

The reviewers agree that there are two interesting aspects of the study.

1) The in vivo demonstration that mitochondrial motility decreases in developing dendrites as activity increases and that this correlation holds for the effects of TTX on motility.

2) The in vitro finding that synaptic activity, rather than spikes, are what matters. We think if both these points were made convincingly, the impact of the study would be appropriate for *eLife*. However, the reviewers think these claims were not fully supported by the data.

The reviewers think the following needs to be done:

(a) cleaning up the analysis of the effects of local events on passing mitochondria to take into account of low overall motility and clarify effects at short intervals;

(b) showing that local calcium transients occur in vivo and have the same effect on mitochondrial motility as they do in vitro;

and

(c) locally activating synapses (e.g., glutamate uncaging or puffing) to show that this is indeed what drives mitochondrial arrests. To be clear -- (a) is required and either (b) or (c) should be sufficient.

Below is a consolidation of the original comments from individual reviewers that led to the above discussion and decision-- in some cases it may seem repetitive with the above, but it may be helpful to see how the reviewers struggled with some analysis as presented.

Summary:

In this study the authors explore the relationship between neural activity and mitochondrial motility in the dendrites of cortical neurons during development. The authors conduct simultaneous imaging of calcium and fluorescently tagged mitochondria motion both in vivo and in an organotypic slice preparation. They show that there is an increase in the frequency of global calcium transients with age and a reduction in motility. However, there is essentially no correlation between motility and spontaneous global calcium transients on a dendrite-by-dendrite level. Rather, they argue that mitochondria motility is influenced by synaptic activity. The data to support this are two-fold: first, mitochondria are more likely to stall near synapses if those synapses have been recently active; second, latrotoxin (which induces exocytosis) but not TTX leads to complete cessation of mitochondria movement. Finally, the authors construct a model to simulate how changes in synaptic activation impact motility making a few assumptions of underlying the arrest duration of mitochondria. The model suggests that given the observed local effects of synaptic activity on mitochondrial movements, the developmental decline in mitochondrial motility could be accounted for by the simultaneous increase in the density of synapses.

Essential revisions:

1. Given there appear to be very few motile mitochondria for any given dendrite, the authors need to be careful as to how they quantify their data. The example shown in Figure 1 appears to have perhaps 1 or 2 motile mitochondria our of 8-10. The quantification of the data is “percent of motile mitochondria”. As seen in Figure 1D, this corresponds to “bins” of percentile changes of 5-10%. Yet, the entire range used to establish the correlation in Figures 2A and 2B is 12%! Hence there is not much confidence that the resulting correlation is meaningful. There is even less confidence that the 1% change in percent observed in TTX (Figure 2F).

2. In figure 2 the authors quantify global Ca events and mitochondrial motility in dendrites in vivo over a range of postnatal days. As has been demonstrated by others (e.g. Faits et al., 2016), the authors see a progressive decrease in motility of mitochondria. They further demonstrate a negative correlation between global Ca events and mitochondria motility. The authors do not present whether they are able to detect spine-specific spontaneous events in-vivo and how spine-specific events change during this period of development. This seems important given the distinction made in vitro.

3. Figure 4 is quite critical to the study but several aspects were confusing. The authors argue that mitochondria are halted near a synapse after the synapse was active. This quantification depends on the length scale that means “near” and the time scale that means “after”. The authors need to clarify this quantification much more. Some questions:

– Figure 4E is based on 2 microns and 120 seconds compared to “before”. Does “before” mean less than 120 seconds or compared only to the time prior to spine activation?

– Figure 4F: the terms of the bootstrap analysis need to be clearly stated – is the hypothesis that seeing a reduction in motility >120 seconds is more than you would expect by chance if all the time points between 0 and 120 seconds are included?

– Figure 4G: I am quite confused here. Let’s take the lightest pink plot. Does this mean if you look at the interval 20 second after the synaptic activation that there are more mitochondria stopping prior to calcium transient than after?

Given all of these questions, the authors must justify 120 seconds as the most relevant time scale. Particularly if they get an opposite sign effect if you look at 20 seconds!

4. The primary manipulation in the paper is the application of LTX in the presence of TTX. This manipulation demonstrates that release of neurotransmitter in the absence of Aps can induce the stopping of mitochondria. However, it seemed unsatisfying that this manipulation was global in nature and not more local (e.g glutamate uncaging/ glutamate puffing/ stimulation of local axons) given that the authors make the distinction earlier between global and local calcium measurements. The authors discuss the potential mechanism by which a local (synaptically induced) calcium transient and a global (backpropagating AP induced) Ca transient could differentially regulate mitochondrial trafficking briefly I the discussion. Mechanistic findings of these differences would certainly elevate our understanding and the paper.

5. The latrotoxin effect is quite dramatic. Though it is true that latrotoxin induces exocytosis, my understanding is that latrotoxin does this causing a massive increase in intracellular calcium and influx of water. Hence latrotoxin may impact mitochondrial motility in a manner independent of synaptic release. Given that the TTX is also likely to impact synaptic release, this seems like the most likely explanation.

6. The authors argue synaptic activity not global cellular activation stops mitochondria. Hence TTX has a small effect and latrotoxin has a big one. But TTX also impact synaptic events as well as global calcium events. So why is there not a bigger impact of TTX on mitochondria? Do they authors argue that most of the synaptic activity is independent of evoked release? This point can be clarified.

7. In Figure 4, the authors present data on the effects of local calcium transients on mitochondrial motility. Panel G indicates that motility is enhanced shortly after the calcium transient and decreases after longer time intervals. This observation appears to approach or reach statistical significance. The authors should clarify whether this is a consistent observation and discuss what mechanisms may account for it.

8. It is not clear for many of the figures (e.g. Figure 2, Figure 3) what the size/ content of the dataset that is being analyzed. In particular, how many mitochondria are being tracked from how many dendrites from how many neurons from how many slices/animals. In figure 4 the text reads “In nine cells (P5 + 3‐7 DIV), we identified 157 spines of which 140 71 (45%) showed spontaneous synaptic calcium transients(376 transients).” This was very helpful to the reader to give an idea of the dataset and it would be helpful to include similar statements for other datasets. This is particularly important given that much of the data is presented as normalized data (e.g. percent moving mitochondria).

9. The authors develop a model that suggests that the local stopping of mitochondria in response to synaptic activity can in large part account for the age-dependent decline in mitochondrial mobility observed in vivo (~70%). The authors’ model suggests that this is only true if the mean arrest duration of mitochondria is around 5 minutes. The data in Figure 4I suggests that the mean arrest duration of mitochondria is about 1 min, but as the authors point out, this is likely an underestimate due to the fact that ~8/20 mitochondria remain at rest when their imaging session ended. Given the importance of this parameter in their model, longer imaging sessions would be necessary to determine mean arrest time more accurately. The data looks like in fact there may be a multimodal distribution of mitochondrial arrest time. As it stands, I don’t feel that the model provides much additional understanding.

[Editors’ note: further revisions were suggested prior to acceptance, as described below.]

Thank you for resubmitting your work entitled "Activity-dependent regulation of mitochondrial motility in developing cortical dendrites" for further consideration by *eLife*. Your article has been reviewed by 3 peer reviewers, one of whom is a member of our Board of Reviewing editors, and the evaluation has been overseen by Gary Westbrook as the Senior Editor.

The manuscript has been greatly improved and two the reviewers felt their concerns were adequately addressed. There remains a major issue regarding the quantification of the "%-motile" data brought up by Reviewer #1 that has not been satisfied. After discussion with all the reviewers it was deemed that this was something that needs to be addressed. We suggest that you consult with a statistician concerning this issue. We will not be able to make a final decision until that issue is adequately addressed.*Reviewer #1:*

Most of my concerns have been addressed in the revision. There is still one major concern remaining – and I have repeated it here.

The remaining concern was the first one I raised in first review and had to do with quantification of % motile mitochondria. Repeating what I said in this first review --

The example shown in Figure 1 appears to have perhaps 1 or 2 motile mitochondria out of 8-10. The quantification of the data is "percent of motile mitochondria". As seen in Figure 1D, this corresponds to "bins" of percentile changes of 5-10%. Yet, the entire range used to establish the correlation in Figures 2A and 2B is 12%! Hence there is not much confidence that the resulting correlation is meaningful. There is even less confidence that the 1% change in percent observed in TTX is meaningful (Figure 2F).

The only answer offered by authors is that this is the same method used in a previous study (MacAskill et al., 2009) but with shorter time windows. However, this is not a question of methods, this is a question of statistics and how reliable the effects are. As noted by authors in this manuscript, the longer time windows in the previous study led to a larger range of percent motile mitochondria, ranging from 0-50%. However this larger range is less susceptible to discretization errors, which is my concern here.

At a minimum, comparisons across conditions (Figure 2) should not be based on t-tests, which assume a normal distribution around a mean. With discretized date like this, the assumption of a normal distribution is incorrect – evidence of this can be seen in Figure 2B where the variance at P8 and P10 actually goes to zero.

An example of the implications of this is Figure 2F – their claim that TTX effects motility in vivo. They find the percent motile goes from 2% to 3% – again a finding based on individual measurements that are binned at >5%. Though they find this incredibly small effect significant using a t-test, this sort of small effect is susceptible to discretization errors.

The authors are requested to perhaps use longer time periods for their session so like the previous paper, it won't as susceptible to this error. At a minimum, the authors should perform a Fisher's exact test rather than a t-test to make statements regarding whether their effects on motility are significant.

*Reviewer #2:*

The authors have satisfactorily addressed my previous concerns.

*Reviewer #3:*

The authors have responded to my concerns regarding the clarity of the paper and analysis. They have addressed my concern regarding the lack of a local manipulation with new experiments (glutamate puffing). This led to a surprising finding that they discuss. Other concerns were addressed through more careful discussion in the manuscript. I feel the manuscript is suitable for publication.

---

## [Author Response]

Essential revisions:1. Given there appear to be very few motile mitochondria for any given dendrite, the authors need to be careful as to how they quantify their data. The example shown in Figure 1 appears to have perhaps 1 or 2 motile mitochondria our of 8-10. The quantification of the data is “percent of motile mitochondria”. As seen in Figure 1D, this corresponds to “bins” of percentile changes of 5-10%. Yet, the entire range used to establish the correlation in Figures 2A and 2B is 12%! Hence there is not much confidence that the resulting correlation is meaningful. There is even less confidence that the 1% change in percent observed in TTX (Figure 2F).

Our analysis is essentially identical to those previously established and accepted (e.g. MacAskill et al., 2009). The only difference is that we determine the percentage of mitochondria for a relatively short interval (bin, 1 second) compared to MacAskill (2 minutes). We intentionally chose 1 second, because it allowed us to compare mitochondrial motility across experimental conditions and for any durations, which was important to infer the exact temporal characteristics of the relationship between synaptic transmission and mitochondrial motility (New Figure 3). As a consequence, our motility rates are lower than reported in previous studies. To confirm that our analysis is equivalent to previous analyses we reanalyzed some of our data as described by MacAskill et al. and found that we get roughly 3x higher values for mitochondrial motility when we chose a 2-minute interval, and these values are in line with previously reported mitochondrial motility rates. Thus, we feel that we use an established analysis approach and we are convinced that this approach allows comparing mitochondrial motility across different conditions, in particular considering that we tracked large numbers of mitochondria (we present these numbers now for all data, see also Point 8 below).

2. In figure 2 the authors quantify global Ca events and mitochondrial motility in dendrites in vivo over a range of postnatal days. As has been demonstrated by others (e.g. Faits et al., 2016), the authors see a progressive decrease in motility of mitochondria. They further demonstrate a negative correlation between global Ca events and mitochondria motility. The authors do not present whether they are able to detect spine-specific spontaneous events in-vivo and how spine-specific events change during this period of development. This seems important given the distinction made in vitro.

We indeed observe local calcium transients in spines in vivo as we do in slices. We mention this now in the Results section and show examples in our new Supplementary Figure 2. Unfortunately, however, we detect these events very rarely, probably because of the lower signal-to-noise ratio and higher incidence of movements in the in vivo recordings. Therefore, we cannot make statements about the relationship between mitochondrial motility and synaptic signaling or the development of synaptic activity in vivo. From a different study in our lab that we prepare for publication currently, we know that synaptic activity increases dramatically during the second postnatal week in vivo. Combined voltage-clamp and calcium imaging recordings at different ages in vivo show that the number of transmission events along dendrites increases 16 times during the second postnatal week. We reference this study in the revised manuscript.

3. Figure 4 is quite critical to the study but several aspects were confusing. The authors argue that mitochondria are halted near a synapse after the synapse was active. This quantification depends on the length scale that means “near” and the time scale that means “after”. The authors need to clarify this quantification much more. Some questions:

We are sorry that the description of our analysis has been confusing. We added new analyses, expanded their description, and made the figure more intuitive. We hope that the reviewers will find this section much clearer now. We addressed the specific points as follows:

– Figure 4E is based on 2 microns and 120 seconds compared to “before”. Does “before” mean less than 120 seconds or compared only to the time prior to spine activation?

For the analysis shown in this panel (now Figure 3I) we compared the percentage of stopping mitochondria during a 120 second interval after a calcium transient occurred with the percentage of stopping mitochondria during a 120 second interval before the transient. When we compare intervals of different durations (e.g. 80, 100, 120 seconds) we always compare an interval of the same duration before and after the occurrence of a calcium transient. We have made this clearer now in the text and by labeling interval durations as “< = interval” within the Figure to indicate that this includes the entire period from the occurrence of a calcium transient until the end of that period.

– Figure 4F: the terms of the bootstrap analysis need to be clearly stated – is the hypothesis that seeing a reduction in motility >120 seconds is more than you would expect by chance if all the time points between 0 and 120 seconds are included?

We performed this bootstrap analysis to test whether the effect size that we observed in the previous panel (previously Figure 4E, now 3I), is higher than one would expect by chance. To test this, we randomized the time points at which local calcium transients occurred and determined the effect size for 1000 randomized data sets. The result shows that the actual effect size is above the 95 percentile of the randomizations, indicating that this is not likely to occur by chance. We have made this clearer in the manuscript now.

– Figure 4G: I am quite confused here. Let’s take the lightest pink plot. Does this mean if you look at the interval 20 second after the synaptic activation that there are more mitochondria stopping prior to calcium transient than after?

We are sorry that this plot was confusing. We have updated it and hope it is clearer now. As the reviewer points out the most confusing aspect was that the 20 second line appeared to suggest a reduction in mitochondrial arrest for this short interval after the occurrence of a calcium transient. This decrease was not significant and was actually based on a very low number of observations (< 10, bottom row in Figure 3 supplement 2, Figure B). Therefore, we removed this line from the plot. We believe that the effect ramps up until pprox.. 1 minute after a calcium transient and becomes more significant with somewhat longer intervals since they include more observations (see Figure 3 supplement 2).

Given all of these questions, the authors must justify 120 seconds as the most relevant time scale. Particularly if they get an opposite sign effect if you look at 20 seconds!

We hope that our new manuscript and the answers to the reviewers’ questions above help making clearer how we derived the spatio-temporal characteristics of mitochondrial arrest from our data. In particular, we find that mitochondria stop moving within 2 minutes after a synaptic calcium transient occurs, probably as a consequence of a second messenger cascade that operates in this time frame. The effect becomes more significant with increasing interval duration (up to 2 minutes) because more observations are included. Therefore, we started with the 120 second interval at short distances to show that mitochondrial arrest is robust and statistically significant in panels I and J of our new Figure 3 and then moved ahead to characterize the spatio-temporal profile of the effect in panels K, L and M of the same Figure.

4. The primary manipulation in the paper is the application of LTX in the presence of TTX. This manipulation demonstrates that release of neurotransmitter in the absence of Aps can induce the stopping of mitochondria. However, it seemed unsatisfying that this manipulation was global in nature and not more local (e.g glutamate uncaging/ glutamate puffing/ stimulation of local axons) given that the authors make the distinction earlier between global and local calcium measurements. The authors discuss the potential mechanism by which a local (synaptically induced) calcium transient and a global (backpropagating AP induced) Ca transient could differentially regulate mitochondrial trafficking briefly I the discussion. Mechanistic findings of these differences would certainly elevate our understanding and the paper.

The reviewer asks for additional evidence that the observed effect of synaptic transmission on mitochondrial arrest is local and a more in-depth analysis of the mechanisms underlying this phenomenon. We have now performed the experiments the reviewer proposed: we puffed glutamate locally and measured both the postsynaptic increase in calcium triggered by glutamate as well as the effect on mitochondrial motility. We observed that glutamate increased the intracellular calcium concentration locally; however, we find only a small, insignificant decrease in mitochondrial motility during a 2 minute period after local glutamate puffing. This finding indicates that glutamate is not causing release-mediated arrest, at least not in the absence of other factors. Thus, we have constrained the possible mechanisms, but unfortunately, could not identify the main factor that mediates mitochondrial arrest at synapses. In the new manuscript we describe a potential mechanism for synaptic transmission induced mitochondrial arrest based on these findings in the discussion.

5. The latrotoxin effect is quite dramatic. Though it is true that latrotoxin induces exocytosis, my understanding is that latrotoxin does this causing a massive increase in intracellular calcium and influx of water. Hence latrotoxin may impact mitochondrial motility in a manner independent of synaptic release. Given that the TTX is also likely to impact synaptic release, this seems like the most likely explanation.

We agree with the reviewer’s notion that the mechanism of latrotoxin induced release of synaptic vesicles is not entirely clear. However, latrotoxin’s action is highly restricted to presynaptic terminals and very specific to synaptic vesicle release. First, the receptors of latrotoxin (latrophilin/CIRL and neurexins) are specifically localized to presynaptic terminals (Valtorta et al., 1984). Furthermore, at the low concentration we used in these experiments (1 nM), latrotoxin triggers vesicle release independently of calcium and affects exclusively readily releasable synaptic vesicles and the reserve pool, but no other cellular organelles (Südhof, 2001) and it does not trigger release of other release-competent organelles, such as dense core vesicles (Matteoli et al., 1988). Therefore, presynaptic vesicle release most likely mediates the strong effect of latrotoxin on dendritic mitochondria. As the reviewer points out, TTX will diminish synaptic release as a consequence of the absence of action potential firing and thus, our observation that TTX increases mitochondrial motility in vivo (Figure 2F, G) is in perfect agreement with our interpretation that vesicle release triggers mitochondrial arrest.

6. The authors argue synaptic activity not global cellular activation stops mitochondria. Hence TTX has a small effect and latrotoxin has a big one. But TTX also impact synaptic events as well as global calcium events. So why is there not a bigger impact of TTX on mitochondria? Do they authors argue that most of the synaptic activity is independent of evoked release? This point can be clarified.

The reviewer wonders why TTX, which, as they point out, affects not only action potential firing, but also reduces synaptic vesicle release, has a relatively small effect on mitochondrial motility, whereas latrotoxin blocks it entirely. TTX and latrotoxin affect vesicle release in opposite directions: TTX leads to a reduction of release (but not an entire block as the reviewers note) and increases mitochondrial motility in vivo by 60% (Figure 2F), whereas latrotoxin stimulates vesicle release and blocks motility entirely in slices. The effect size of TTX should depend on the level of spontaneous activity in a given cell before TTX application: if activity is low and mitochondrial motility mostly unhindered by synaptic blockade, TTX will have a small effect. Conversely, when activity levels are high, TTX should have a large effect. This is exactly what we observe (Figure 2G). Furthermore, the 60% increase in mitochondrial motility after TTX application in vivo is perfectly in line with the estimate of our model for the percentage of moving mitochondria that halt in response to synaptic activity. In slice cultures, we see a smaller effect of TTX. This is probably a consequence of the fact that the slice cultures were a bit less mature and showed lower levels of spontaneous activity.

7. In Figure 4, the authors present data on the effects of local calcium transients on mitochondrial motility. Panel G indicates that motility is enhanced shortly after the calcium transient and decreases after longer time intervals. This observation appears to approach or reach statistical significance. The authors should clarify whether this is a consistent observation and discuss what mechanisms may account for it.

As pointed out above, we agree with the reviewers that our original Figure 4G was confusing. The line for the 20 second interval indeed appeared to indicate that mitochondrial motility was increased during this interval. However, this is not the case: we do not find a significant effect for this interval and the number of observations for this interval was very low, indicating that the downward deflection was spurious. To avoid the false impression that mitochondrial motility is increased for short intervals we have removed the 20 second interval data from this figure (New Figure 3K).

8. It is not clear for many of the figures (e.g. Figure 2, Figure 3) what the size/ content of the dataset that is being analyzed. In particular, how many mitochondria are being tracked from how many dendrites from how many neurons from how many slices/animals. In figure 4 the text reads “In nine cells (P5 + 3‐7 DIV), we identified 157 spines of which 140 71 (45%) showed spontaneous synaptic calcium transients(376 transients).” This was very helpful to the reader to give an idea of the dataset and it would be helpful to include similar statements for other datasets. This is particularly important given that much of the data is presented as normalized data (e.g. percent moving mitochondria).

We have added the number of experiments, animals and mitochondria tracked to the figure legends. In addition, we made a new supplemental Figure (sFigure 4, Figure in response to Point 3, above) to give an overview of the number of observations for each distance/interval bin in our new Figure 3K, L (previously 4G, H).

9. The authors develop a model that suggests that the local stopping of mitochondria in response to synaptic activity can in large part account for the age-dependent decline in mitochondrial mobility observed in vivo (~70%). The authors’ model suggests that this is only true if the mean arrest duration of mitochondria is around 5 minutes. The data in Figure 4I suggests that the mean arrest duration of mitochondria is about 1 min, but as the authors point out, this is likely an underestimate due ’o the fact that ~8/20 mitochondria remain at rest when their imaging session ended. Given the importance of this parameter in their model, longer imaging sessions would be necessary to determine mean arrest time more accurately. The data looks like in fact there may be a multimodal distribution of mitochondrial arrest time. As it stands, I don’t feel that the model provides much additional understanding.

The reviewer points out that we modeled the effect of synaptic activity on mitochondrial motility for several mean arrest durations and that we find a maximal effect of 70% at a mean arrest duration of 5 minutes. In fact, we believe that this is an upper bound of the effect. The real mean arrest duration is between 1 and 5 minutes and we estimate in the manuscript that the increase in synaptic transmission between the first and the second postnatal week would reduce mitochondrial motility by 30-60%. Since we find mitochondrial motility to be reduced by 70% between the first and the second postnatal week, our model suggests that 43-86% of the developmental effect can be explained by synaptic transmission induced arrest. Thus, in our mind the model demonstrates biological relevance of this mechanism for the frequently observed, but so far unexplained, observation that mitochondrial motility decreases during development. The reviewer argues that it would be necessary to image for longer periods of time to get an even more precise estimate of the real arrest duration. Unfortunately however, we are not comfortable having longer time-lapse recordings since we need to avoid any risk of photo-damage on the cells. Finally, the reviewer suggests that there is a bimodal distribution of arrest durations. While we believe that this could indeed be a possibility, we find that a unimodal distribution as plotted in Figure 5A can look like a bimodal distribution, simply because of the constraint that there are no negative arrest durations, causing a relatively high peak in the lowest bin of the histogram. This is now stated more clearly in the text.

[Editors’ note: further revisions were suggested prior to acceptance, as described below.]

Reviewer #1Most of my concerns have been addressed in the revision. There is still one major concern remaining – and I have repeated it here.The remaining concern was the first one I raised in first review and had to do with quantification of % motile mitochondria. Repeating what I said in this first review --The example shown in Figure 1 appears to have perhaps 1 or 2 motile mitochondria out of 8-10. The quantification of the data is "percent of motile mitochondria". As seen in Figure 1D, this corresponds to "bins" of percentile changes of 5-10%. Yet, the entire range used to establish the correlation in Figures 2A and 2B is 12%! Hence there is not much confidence that the resulting correlation is meaningful. There is even less confidence that the 1% change in percent observed in TTX is meaningful (Figure 2F).The only answer offered by authors is that this is the same method used in a previous study (MacAskill et al. 2009) but with shorter time windows. However, this is not a question of methods, this is a question of statistics and how reliable the effects are. As noted by authors in this manuscript, the longer time windows in the previous study led to a larger range of percent motile mitochondria, ranging from 0-50%. However this larger range is less susceptible to discretization errors, which is my concern here.At a minimum, comparisons across conditions (Figure 2) should not be based on t-tests, which assume a normal distribution around a mean. With discretized date like this, the assumption of a normal distribution is incorrect – evidence of this can be seen in Figure 2B where the variance at P8 and P10 actually goes to zero.An example of the implications of this is Figure 2F – their claim that TTX effects motility in vivo. They find the percent motile goes from 2% to 3% -- again a finding based on individual measurements that are binned at >5%. Though they find this incredibly small effect significant using a t-test, this sort of small effect is susceptible to discretization errors.The authors are requested to perhaps use longer time periods for their session so like the previous paper, it won't as susceptible to this error. At a minimum, the authors should perform a Fisher's exact test rather than a t-test to make statements regarding whether their effects on motility are significant.

We have now addressed the concern of Reviewer 1 as follows:

We agree with the reviewer that the statistics we used for comparisons in Figure 2 do not entirely rule out the possibility that the effects we observe are spurious. In consultation with Dr. Matthew Self, an expert in biomedical statistics, we reanalyzed the relevant data in a way that allowed us to follow the advice of Reviewer 1. We summed the number of moving versus stable mitochondria for every 2 minute bin from all recordings during the respective conditions to generate a contingency table (see Author response table 1). We performed a Fisher’s Exact test on these contingency tables. We find that there are indeed highly significant differences between the investigated conditions. We feel thus, that the new analyses support our conclusions that (1) mitochondrial motility decreases with age and that (2) TTX increases mitochondrial motility significantly. We added these analyses and the statistical tests to the Methods section as part of an extended discussion on the statistics. In the new manuscript, we still show the %-moving mitochondria values, because they can be compared directly to all other measurements in the manuscript, but refer the reader to the Statistics section in Materials and methods. In addition, we include the respective contingency tables in the Source Data Tables. We hope that the reviewers and editors agree with us that the new analyses clearly support our conclusions regarding the factors that regulate mitochondrial motility in vivo.

**Author response table 1. sa2table1:** Contingency table related to data shown in Figure 2F. The Fisher’s exact test statistic value is 0.0034.

	Moving mitochondria	Stable mitochondria
Baseline	70	599
TTX	172	946